# Knowledge of Polish Nurses About Sepsis Based on Validated Questionnaire: A Multi-Site Cross-Sectional Study

**DOI:** 10.3390/nursrep15060195

**Published:** 2025-05-30

**Authors:** Nicole Bartulewicz, Lena Serafin, Bożena Czarkowska-Pączek

**Affiliations:** Department of Clinical Nursing, Health Sciences Faculty, Medical University of Warsaw, Erazma Ciołka Street 27, 01-445 Warsaw, Poland; n.bartulewicz@gmail.com (N.B.); bpaczek@wum.edu.pl (B.C.-P.)

**Keywords:** sepsis, nursing, knowledge, survey, education, nurse

## Abstract

**Background**: Nurses play a fundamental role in identifying the early symptoms of sepsis and thereby contribute to early diagnosis and prevention, which decreases complications and mortality rates and lowers the cost of care. This study aimed to evaluate nurses’ knowledge of sepsis and to analyze its relationship with attitude, self-assessment, and sociodemographic variables. **Methods**: A correlational, cross-sectional study was conducted. A total of 293 nurses in Poland using a validated tool, the Nurses’ Attitudes and Knowledge about Sepsis Scale (NAKSeS), which assesses both knowledge and attitudes toward sepsis. The instrument included two knowledge subscales (Factor 1: knowledge of pathogenesis, treatment, and prevention; Factor 2: knowledge of nursing actions), an attitude subscale, and a self-assessment item. Sociodemographic data included age, seniority, voluntary postgraduate education, current workplace, and place of residence. **Results**: Nurses demonstrated moderate levels of general knowledge, Factor 1, Factor 2, attitude toward sepsis, and self-assessed knowledge. Higher scores across all domains were observed among nurses who had completed postgraduate education, cited professional experience or books as key sources of knowledge, and worked in high-acuity settings such as intensive care units, emergency departments, or pediatric wards. Nurses working in larger cities scored significantly higher in general knowledge and Factor 1 compared to those in smaller towns or rural areas. Additionally, greater age and longer work experience were positively associated with more favorable attitudes and higher self-assessed knowledge, although negatively correlated with some knowledge scores. **Conclusions**: Nurses’ knowledge and attitudes toward sepsis were influenced by the source of education and clinical exposure, with significantly better outcomes observed among those with postgraduate training and experience in high-acuity settings. These findings underscore the need to strengthen sepsis education across all levels of nursing curricula and promote accessible, continuous professional development supported by validated assessment tools.

## 1. Introduction

The search for the appropriate definition of sepsis has accelerated due to the increasing number of morbidities and fatalities, as well as the WHO (World Health Organization) resolution that obliges countries to raise awareness about sepsis [1]. It is estimated that there are around 50 million sepsis cases annually, with 11 million of those being fatal; this represents 20% of all global deaths [2]. Therefore, since 2016, sepsis has been defined as a life-threatening organ dysfunction caused by a dysregulated response of the host to infection, with a rejection of the definition of SIRS (systematic inflammatory response syndrome) and a narrowing of the guidelines [3,4,5].

In Poland, although comprehensive national data on sepsis prevalence are limited, local hospital reports indicate a growing number of cases, particularly in emergency departments and intensive care units. These trends are consistent with global patterns and underline the need to strengthen sepsis-related competencies among healthcare workers, especially nurses [6,7]. There are no national guidelines regulating sepsis education for nurses, and formal postgraduate education is voluntary. As a result, the level of knowledge and attitudes among nurses remains highly variable, with educational gaps persisting across clinical settings.

Considering the above, proper preparation by healthcare providers regarding sepsis awareness appears to be crucial in delivering high-quality care. Evaluating their knowledge and attitudes toward sepsis will allow for the designing of professional development interventions to ensure good care outcomes.

This study aimed to evaluate Polish nurses’ knowledge of sepsis in three domains—(a) knowledge of pathogenesis, methods of treatment, and prevention; (b) knowledge of nursing activities undertaken; and (c) attitudes toward sepsis. We also analyzed how these domains relate to self-assessment and selected sociodemographic variables, using a validated instrument developed specifically for this purpose [8].

## 2. Background

To appropriately apply the new definition of sepsis, adhere to guidelines, and implement the WHO resolution, it is essential to first acquire knowledge about sepsis—including screening methods, procedures to follow when sepsis is suspected, and the risks associated with delays in appropriate medical intervention and treatment. The definition of sepsis is based on organ dysfunction, which can be assessed by the SOFA (sequential organ failure assessment) scale. The scale contains clinical features meant to draw healthcare providers’ attention: impaired oxygenation; jaundice, cholestasis; reduced GFR (glomerular filtration rate), reduced urine output; and ventricular dilatation, reduced ejection fraction, and reduced heart contractility [9]. Morbidity and mortality contribute equally to a rising SOFA score. Other clinical variables and tools used for sepsis screening include systemic inflammatory response syndrome criteria, the National Early Warning Score (NEWS), and the Modified Early Warning Score (MEWS) [5]. Because not all units can obtain quick laboratory values, which are necessary for the SOFA scale, the quick SOFA (qSOFA) was developed. Noninvasive variables—systolic blood pressure, respiratory rate, and mental status—are taken into consideration by using qSOFA. However, according to the last revision of the international guidelines for the management of sepsis and septic shock, issued in 2021, [5] qSOFA has rather poor sensitivity and should not be implemented as a single screening tool. On the other hand, although only about 24% of patients with infection have a qSOFA score of 2 or 3, these patients have the worst prognosis and account for 70% of poor outcomes [5]. Therefore, the presence of a positive qSOFA should always alert the healthcare provider to the possibility of sepsis and the need for intensive care unit (ICU) transfer. Appropriate screening can improve patient outcomes by initiating the appropriate reaction of the medical team at the right time [10,11]. The research shows that nearly 70% of sepsis cases are community-acquired, which draws attention to the proper education of patients by medical personnel [1].

Although diagnostic tools are available, successful early recognition of sepsis depends largely on healthcare providers’ awareness, attitudes, and vigilance. Nurses, in particular, are crucial in this process, as they are often the first to notice changes in a patient’s condition and initiate emergency procedures [10,11]. Previous research highlights knowledge deficits among nurses regarding early symptoms, pathophysiology, and evidence-based interventions in sepsis [12,13,14]. Attitudes toward sepsis—including perceived competence, emotional readiness, and willingness to act—are also suboptimal in many settings [15]. These gaps are especially concerning given the crucial role of nurses in initiating time-sensitive interventions.

Some studies show that participation in targeted sepsis education significantly improves both knowledge and confidence levels among nurses [16,17,18]. However, the availability and uptake of such programs are limited. In Poland, formal postgraduate education is not mandatory and varies by region, leading to uneven levels of preparation. This creates an urgent need to assess baseline knowledge and attitudes to inform tailored educational strategies.

Tools used to assess sepsis-related competence among nurses include both ad hoc surveys and validated scales. However, few of them combine knowledge and attitudes in one instrument. This study used the NAKSeS (Nurses’ Attitudes and Knowledge about Sepsis Scale), a validated tool designed to assess both cognitive and affective domains in a single framework [7].

Evaluating nurses’ knowledge and attitudes toward sepsis can support the development of more effective, evidence-based educational interventions aimed at improving sepsis-related outcomes.

## 3. The Study

### 3.1. Aims

This study aimed to evaluate nurses’ knowledge of sepsis in three domains: (a) knowledge of pathogenesis, methods of treatment, and prevention; (b) knowledge of nursing activities undertaken; and (c) attitude. Also, an additional goal was to analyze the relationship between the level of knowledge with attitude, self-assessment, and their correlation with sociodemographic variables.

### 3.2. Design

A correlational, cross-sectional study was conducted. The study was a self-reported electronic survey. A link to the questionnaire was shared by websites associated with the nursing community and also by medical universities in Poland, which spread the link to their second-cycle students, who concomitantly serve as practicing nurses. The STROBE (Strengthening The Reporting of OBservational Studies in Epidemiology) reporting guidelines were used in both the framing and reporting of this study.

### 3.3. Participants

The nurses were selected using the convenience sampling method. To be included in the study, nurses had to have been working as nurses in primary healthcare or the hospital ward for at least three months. The three-month threshold was adopted as it typically marks the completion of initial onboarding and the beginning of independent clinical work, including exposure to sepsis-related situations. Exclusion criteria were nurses who worked in administration, management, education, and other non-patient-care areas and who had been practicing for less than three months. All nurses who took part in the study were living and working in Poland. To reduce sampling bias, we ensured access was available to nurses working in various settings (e.g., hospitals, primary care) across different regions of Poland. To ensure a statistical power of at least 0.80, medium effect size, confidence interval of 0.95, and α of 0.05, the analysis required at least 138 nurses. This calculation was performed a priori via the software G*Power 3.1.9.7. In total, 332 questionnaires were collected, from which 293 (88%) fit the inclusion criteria.

### 3.4. Data Collection

The study was carried out in 2020. Data collection was conducted using the NAKSeS (Nurses’ Attitudes and Knowledge about Sepsis Scale), which was developed by the authors of this study [7].

The NAKSeS is a 23-item self-report instrument composed of three subscales: (1) knowledge of pathogenesis, treatment methods, and prevention (11 items); (2) knowledge of nursing actions in suspected sepsis (6 items); and (3) attitude toward sepsis (6 items). The knowledge subscales were measured using a 3-point Likert-type format (“yes”, “no”, “I don’t know”). Correct answers were awarded 1 point; incorrect and “I don’t know” responses were scored 0. In total, the maximum possible score was 17: 11 points for Factor 1 and 6 points for Factor 2.

The attitude subscale included 6 statements rated on a 5-point Likert scale (1 = strongly disagree, 2 = disagree, 3 = neither agree nor disagree, 4 = agree, 5 = strongly agree). All statements were positively worded; therefore, reverse scoring was not necessary. The total score for this subscale ranged from 6 to 30 points. The self-assessment consisted of one item asking nurses to rate their own level of knowledge about sepsis on a 5-point Likert scale (1 = very low, 5 = very high).

Higher scores on the scale indicate a greater level of knowledge. On the scale concerning the assessment of knowledge, nurses received a point for each correct answer and no point for an incorrect answer or an “I don’t know” answer. Thus, the total possible points were as follows: 17 points for general knowledge, 11 points for Factor 1, 6 points for Factor 2, and 30 points for attitude as measured on the Likert scale. In terms of self-assessment, a maximum of 5 points could be obtained.

The second part of the questionnaire was the metrics. It contained 9 additional questions regarding the nurses’ sex, age, work experience, level of education, postgraduate qualifications, current workplace, number of patients per nurse in the workplace, and source of knowledge of sepsis.

### 3.5. Ethical Considerations

The study was approved by the Bioethics Committee of the Medical University (AKBE/96/2018) and performed in accordance with the Declaration of Helsinki. All nurses were informed about the purpose of the study, terms and conditions, and voluntary participation. Nurses were unable to begin the survey unless they had given their written consent. Each nurse was assured of anonymity, confidentiality, and the possibility of withdrawing from participation at any time.

### 3.6. Data Analysis

Statistical analyses were performed using IBM SPSS Statistics 25.0. The program was used to calculate basic descriptive statistics together with the Kolmogorov–Smirnov distribution normality test. To compare the two groups in terms of the analyzed variables, the analysis was performed with Student’s *t*-test for independent samples. When there were more than two groups, a one-way analysis of variance was performed. To establish the dependence between the variables, a Pearson or Spearman’s correlation analysis was performed. For the analysis, α = 0.05 was assumed to be the level of significance.

### 3.7. Validity and Reliability

The reliability of the general knowledge subscale in the original version of the scale was KR-20 = 0.718, and for the attitude subscale, Cronbach’s alpha was 0.884 [7].

In the present study, internal consistency was also assessed. For the general knowledge subscale (with dichotomous response options), KR-20 was 0.745. For the attitude subscale, Cronbach’s alpha was 0.857.

## 4. Results

### 4.1. Demographic Data

Of the 293 nurses, 277 (95%) were women. The age of the nurses ranged from 21 to 61 years old, and the overall average age was 37 years. Of the total, 139 (47%) nurses had work experience from a few months to 10 years, and 19 (7%) had practiced 31–40 years. The overall mean of experience was 14 years.

Only 39 (13%) nurses declared that they did not attend postgraduate education. Of the total, 50 (17%) had worked in the department of anesthesiology and intensive care. In total, 101 (35%) worked in a city with over 500,000 residents. Lastly, the most common source of knowledge of sepsis declared by nurses was basic vocational education (n = 191, 65%) and work experience (n = 154, 53%). The sample characteristics are presented in Table 1.

### 4.2. Nurses’ Knowledge and Attitudes Toward Sepsis

Nurses in our study presented an average level of general knowledge (M = 9.49 out of 17). The level of knowledge in Factor 1 (knowledge of pathogenesis, methods of treatment and prevention), Factor 2 (knowledge of nursing activities undertaken), and attitude toward sepsis was also at a moderate level, with mean scores of 5.72 out of 11, 3.77 out of 6, and 21.65 out of 30, respectively. Additionally, the self-assessment of nurses regarding the question of whether they define their level of knowledge as sufficient was tested. To interpret the data, a 5-point Likert scale was used. The medium score of 3.18 indicates that nurses do not feel competent when it comes to sepsis. The percentage of scores of nurses on each scale is presented in Figure 1.

Afterward, the dependence between general knowledge and three factors—(1) knowledge of pathogenesis, methods of treatment and prevention; (2) knowledge of nursing activities undertaken; and (3) attitude—was verified by Pearson’s correlation. The strongest correlation appeared between general knowledge and Factor 1 (r = 0.87, *p* < 0.0010), as well as between self-assessment of knowledge of sepsis and attitude (r = 0.7, *p* < 0.001). The results are shown in Table 2.

### 4.3. Comparison of the Level of General Knowledge, Its Dimensions, Attitude, and Self-Assessment of Knowledge Regarding the Level of Voluntary Postgraduate Education

Participation in specialization training has an impact on nurses’ attitude toward sepsis, *t* = −3.64, *p* < 0.001, *d* = 0.44, 95% *CI* [−3.00, −0.89]. Nurses who participated in specialization training have a higher level of attitude (M = 22.84, SD = 4.35) than those who did not (M = 20.89, SD = 4.52). Also, participation in qualification courses has an impact on nurses’ attitude toward sepsis, *t* = −2.59, *p* = 0.010, *d* = 0.30, 95% *CI* [−2.41, −0.33]. Nurses who participated in qualification courses have a higher level of attitude (M = 22.30, SD = 4.22) than those who did not (M = 20.93, SD = 4.79). Our study also revealed the impact of participation in specialized courses on attitude, *t* = −3.55, *p* < 0.001, *d* = 0.41, 95% *CI* [−2.88, −0.82], and self-assessment of knowledge, *t* = −3.20, *p* = 0.002, *d* = 0.37, 95% *CI* [−0.63, −0.15]. Nurses who completed their specialized course have a higher level of attitude (M = 22.53, SD = 4.39) and higher self-assessment of knowledge (M = 3.37, SD = 1.02) than nurses who did not participate in this kind of course (M = 20.68, SD = 4.53; M = 2.98, SD = 1.06, respectively).

Moreover, not participating in any postgraduate form of education has an impact on the presented level of attitude, *t* = 4.79, *p* < 0.001, *d* = 0.82, 95% *CI* [2.13, 5.10]. Nurses who did not complete any postgraduate form of education present a lower level of attitude (M = 18.51, SD = 5.07) than nurses who participated in some postgraduate education (M = 22.13, SD = 4.27). Not participating in any postgraduate form of education has an impact on nurses’ self-assessment of knowledge, *t* = 3.33, *p* = 0.001, *d* = 0.57, 95% *CI* [0.24, 0.94]. Nurses who did not complete any postgraduate form of education present a lower level of attitude (M = 2.67, SD = 1.08) than nurses who participated in some postgraduate education (M = 3.26, SD = 1.03).

Detailed data are provided in the Appendix A.

### 4.4. Comparison of the Level of General Knowledge, Its Dimensions, Attitude, and Self-Assessment of Knowledge Regarding the Source of Knowledge of Sepsis

The analysis showed that gaining knowledge from undergraduate education has an impact on nurses’ attitude toward sepsis, *t* = 2.31, *p* = 0.022, d = 0.27, 95% *CI* [0.18, 2.29]. Nurses who gained their knowledge from undergraduate education had a lower attitude level (M = 21.21; SD = 4.68) than nurses who declared other sources of knowledge of sepsis (M = 22.45; SD = 4.19). It has also been revealed that gaining knowledge from undergraduate education has an impact on nurses’ self-assessment of level of knowledge, *t* = 2.53, *p* = 0.012, *d* = 0.31, 95% *CI* [0.07, 0.58]. Nurses who gained their knowledge from undergraduate education self-assessed their level of knowledge lower (M = 3.07; SD = 1.06) than people who declared other sources of knowledge of sepsis (M = 3.39, SD = 1).

Moreover, the source of knowledge based on postgraduate education had an impact on nurses’ general knowledge, *t* = −3.39, *p* = 0.001, *d* = 0.48, 95% *CI* [−2.41, −0.64]. Nurses who gained their knowledge from postgraduate education presented a significantly higher level of general knowledge (M = 10.64; SD = 3.04) than nurses who declared other sources of knowledge of sepsis (M = 9.11; SD = 3.4). The source of knowledge based on postgraduate education also had an impact on nurses’ knowledge regarding: Factor 2: *t* = −4.50, *p* < 0.001, *d* = 0.53, 95% *CI* [−1.26, −0.47]. Nurses who gained their knowledge from postgraduate education presented a significantly higher level of knowledge of nursing activities undertaken regarding suspicion of sepsis (M = 4.43; SD = 1.31) than nurses who declared other sources of knowledge of sepsis (M = 3.56; SD = 1.75).

Attitude: *t* = −3.48, *p* = 0.001, *d* = 0.47, 95% *CI* [−3.30, −0.92]. Nurses who gained their knowledge from postgraduate education presented a significantly higher level of attitude (M = 24.24; SD = 4.36) than nurses who declared other sources of knowledge of sepsis (M = 21.13; SD = 4.50).

Self-assessment of knowledge: *t* = −2.34, *p* = 0.020, *d* = 0.32, 95% *CI* [−0.61, −0.05]. Nurses who gained their knowledge from postgraduate education presented a significantly higher level of attitude (M = 3.43; SD = 1.05) than nurses who declared other sources of knowledge of sepsis (M = 3.10; SD = 1.04).

A significantly higher level of general knowledge of sepsis, all of the dimensions of NAKSeS, and self-assessment of knowledge were presented by nurses who gained their knowledge of sepsis from professional experience and books. Detailed data are presented in Table 3.

### 4.5. Comparison of the Level of General Knowledge, Its Dimensions, Attitude, and Self-Assessment of Knowledge Regarding the Age, Professional Experience, and Workplace (City)

The age of nurses was negatively and weakly associated with the level of general knowledge and Factor 1 (r_s_ = −0.12, *p* = 0.042; r_s_ = −0.18, *p* = 0.003, respectively). Additionally, it was weakly and positively related to attitude (r_s_ = 0.24, *p* < 0.001) and self-assessment of knowledge (r_s_ = 0.13, *p* = 0.02).

A similar result arose in correlation with one of the NAKSeS dimensions and nurses’ self-assessment of knowledge and nurses’ professional experience. Professional experience was weakly and negatively correlated with Factor 1 (r_s_ = −0.14, *p* = 0.016) and weakly and positively associated with attitude (r_s_ = 0.27, *p* < 0.001) and self-assessment (r_s_ = 0.15, *p* = 0.009).

Workplace placement in a larger city was positively and weakly related to the level of general knowledge and Factor 1 (r_s_ = 0.20, *p* = 0.001; r_s_ = 0.22, *p* < 0.001, respectively).

Detailed data are presented in Table 4.

### 4.6. Comparison of the Level of General Knowledge, Its Dimensions, Attitude, and Self-Assessment of Knowledge Regarding the Workplace Setting

Working in the department of anesthesiology and intensive care has an impact on general knowledge, *t* = −5.32, *p* < 0.001, *d* = 0.70, 95% *CI* [−3.13, −1.43]; Factor 1, *t* = −3.96, *p* < 0.001, *d* = 0.53, 95% *CI* [−2.00, −0.66]; and Factor 2, *t* = −4.44, *p* < 0.001, *d* = 0.57, 95% *CI* [−1.38, 0.53]. Nurses who worked in the department of anesthesiology and intensive care had a higher level of general knowledge (M = 11.38, SD = 2.61), Factor 1 (M = 6.82, SD = 2.07), and Factor 2 (M = 4.56, SD = 1.30) than those who had not worked in such areas (M = 9.10, SD = 3.39; M = 5.49, SD = 2.58; M = 3.61, SD = 1.72, respectively).

Emergency ward experience also has an impact on general knowledge, *t* = −2.82, *p* = 0.005, *d* = 0.71, 95% *CI* [−4.00, 0.71]; Factor 1, *t* = −2.15, *p* = 0.032, *d* = 0.54, 95% *CI* [−2.61, 0.12]; and Factor 2, *t* = −2.97, *p* = 0.008, *d* = 0.59, 95% *CI* [−1.69, −0.29]. A higher level of general knowledge (M = 11.71, SD = 3.46), Factor 1 (M = 7.00, SD = 3.06), and Factor 2 (M = 4.71, SD = 1.31) was found among nurses who worked in the emergency ward than among those who did not (M = 9.35, SD = 3.33; M = 5.64, SD = 2.50; M = 3.71, SD = 1.70, respectively).

The study showed the impact of working as a pediatric nurse on general knowledge, *t* = −2.72, *p* = 0.011, *d* = 0.31, 95% *CI* [−1.82, −0.26], and Factor 1, *t* = −3.84, *p* = 0.001, *d* = 0.56, 95% *CI* [−2.19, −0.66]. Pediatric nurses presented a higher level of general knowledge (M = 10.47, SD = 1.33) and Factor 1 (M = 7.06, SD = 1.39) than nurses without such experience (M = 9.43, SD = 3.46; M = 5.63, SD = 2.58, respectively).

Working in other units impacted general knowledge, *t* = 4.18, *p* < 0.001, *d* = 0.50, 95% *CI* [0.87, 2.41]; Factor 1, *t* = 3.40, *p* = 0.001, *d* = 0.40, 95% *CI* [0.43, 1.60]; Factor 2, *t* = 3.14, *p* = 0.002, *d* = 0.37, 95% *CI* [0.23, 1.02]; and attitude, *t* = 2.83, *p* = 0.005, *d* = 0.33, 95% *CI* [0.45, 2.50]. Nurses who worked in other units have a lower level of general knowledge (M = 8.83, SD = 3.38), Factor 1 (M = 5.31, SD = 2.55), Factor 2 (M = 3.52, SD = 1.68), and attitude (M = 21.05, SD = 4.76) than those who worked in the department of anesthesiology and intensive care, emergency ward, or pediatric ward (M = 10.47, SD = 3.14; M = 6.32, SD = 2.45; M = 4.14, SD = 1.66; M = 22.53, SD = 4.08, respectively).

Primary healthcare nurses’ knowledge did not show specific differences compared to that of other groups.

Detailed data are provided in the Appendix A.

## 5. Discussion

This study aimed to investigate nurses’ knowledge and attitude toward sepsis according to a new WHO definition. Nurses play a fundamental role in identifying the early symptoms of sepsis, thereby contributing to early clinical diagnosis and prevention, which decreases the complication and mortality rates and lowers the costs of care [19,20]. Therefore, there is a need to constantly monitor their preparation and attitudes toward sepsis. However, assuming the knowledge or competence gaps of nurses is not enough. Coiner and Wingo’s literature review showed that factors related to sepsis knowledge differ in each nursing population; however, sepsis-focused education was the strongest predictor of knowledge and could even compensate for a lack of professional experience in this field [8].

In our study, nurses present an average level of knowledge of sepsis in both objective and subjective assessment. Moreover, the average number of points in the “attitude” dimension confirms the average openness to care about patients with sepsis. Our results are in line with other evidence, which presents both insufficient nurses’ knowledge of sepsis and insufficient knowledge of this issue among other healthcare workers [8,14,15,21,22,23]. Furthermore, Breen and Rees’ study revealed that nursing delays and knowledge deficits are the top barriers leading to delays in sepsis treatment [24]. In Iran, a cross-sectional study among ICU head nurses revealed weak knowledge and practices concerning infection prevention and control measures, despite a positive attitude. This suggests a gap between knowledge and practice, emphasizing the need for enhanced educational interventions [25]. These international findings underscore the importance of tailoring educational programs to address specific knowledge gaps and practice deficiencies. Differences in nursing education systems, such as curriculum content, clinical exposure, and continuing education requirements, may contribute to the variability in sepsis knowledge and management practices among nurses globally.

The following investigation showed that as the self-assessment of the level of knowledge increases, the attitude of nurses toward sepsis—understood as readiness and willingness to care for patients with sepsis—improves. Assuming that the self-assessment of knowledge is grounded in reliable knowledge, its strengthening should be based on additional training in the topic of sepsis, the effectiveness of which has been shown in previous studies, including in terms of improving nurses’ attitude toward sepsis [26]. Educational programs and coaching approaches are needed to increase nurses’ ability to make decisions regarding early recognition, assessment, and intervention for sepsis symptoms [21]. Nevertheless, the consolidation of the actions leading to the acquisition of knowledge should be supported by guidelines that will help with quick recognition and adequate decision-making [27,28]. Knowledge and use of clinical guidelines and sepsis screening tools are established methods to help reduce patient mortality [29].

Nevertheless, our study shows that the highest level of knowledge and attitudes is presented by nurses who learn about sepsis from postgraduate education and professional experience. It should be stressed here that postgraduate education is fully voluntary in Poland. To apply for a professional license, it is enough to graduate from a bachelor’s program, e.g., undergraduate education. In such circumstances, it is very important to note that it is not an undergraduate education that provides nurses with sufficient knowledge of sepsis. In Poland, where postgraduate education is voluntary and access may vary, our study highlights the necessity of integrating comprehensive sepsis education into both undergraduate curricula and continuing professional development programs. Implementing standardized assessment tools, like the NAKSeS, can aid in identifying educational needs and evaluating the effectiveness of training interventions. Harley et al. indicated that there was incomplete preparation of last-year nursing students to recognize, escalate, and manage sepsis [30]. An Italian study demonstrated that nurses and physicians who participated in educational workshops based on the Surviving Sepsis Campaign guidelines showed significant improvements in knowledge and attitudes toward sepsis management. This underscores the importance of continuous education in shaping positive attitudes and effective practices [31]. Emphasizing the importance of practical training in increasing knowledge and attitude, which can also be carried out in simulation centers, is in line with the recommendation from other investigations of analyzing issue [8].

The results regarding knowledge and attitude mentioned above correspond with the finding that nurses from larger cities achieve better scores. Usually, in larger cities, the hospital wards have a higher volume in terms of the number of patients, especially critically ill ones, which corresponds to the better professional experience of nurses. A systematic review conducted by Abdalhafith et al. (2025) showed that the knowledge, confidence, and clinical decision-making skills of intensive care nurses in managing sepsis vary and are often dependent on the availability of training and institutional support. Deficiencies in these areas may be more pronounced in rural settings, where access to educational resources is limited [32]. Also, most knowledge of sepsis results from voluntary postgraduate education, and access to such forms of education is better in larger cities.

Based on the findings regarding the impact of various educational sources on nurses’ knowledge and attitudes toward sepsis, it is evident that the origin of sepsis-related education significantly influences both self-assessed and objectively measured competencies. Nurses who acquired their sepsis knowledge through postgraduate education demonstrated notably higher levels of general knowledge, specific competencies (as measured by Factor 2), attitudes, and self-assessment scores compared to those who relied on undergraduate education or other sources. This aligns with previous studies indicating that postgraduate training and continuous professional development are crucial in enhancing nurses’ proficiency in sepsis recognition and management. For instance, a systematic review highlighted that educational interventions incorporating active learning strategies, such as simulations, significantly improve healthcare professionals’ knowledge and patient outcomes related to sepsis care [33]. Conversely, reliance solely on undergraduate education was associated with lower attitude levels and self-assessed knowledge. This finding is consistent with research conducted among nursing students in Croatia, Cyprus, and Greece, which revealed limited knowledge about sepsis and emphasized the need for enhanced curricular content on sepsis in undergraduate programs [34]. Furthermore, nurses who cited professional experience and self-directed learning through books as their primary sources of sepsis knowledge also exhibited higher levels of general knowledge, specific competencies, attitudes, and self-assessment. This suggests that experiential learning and proactive information seeking contribute positively to nurses’ competence in sepsis care. These findings underscore the importance of fostering a culture of continuous learning and providing accessible resources for self-education.

Our study also showed that with an increase in the age and seniority of nurses, which is also linked to practical experience, the level of attitudes of nurses toward sepsis increases. A study by Rababa et al. (2022) found that nurses with more than five years of experience and those holding a master’s degree reported significantly better knowledge, attitudes, and practices related to sepsis management compared to their less experienced counterparts [15]. According to Hogg and Vaughan, attitude is a relatively enduring organization of beliefs, feelings, and behavioral tendencies [35]. Therefore, to ensure the best patient outcomes, in addition to raising the knowledge of nurses, it is important to develop attitudes that directly affect behavior. Multigene rationality in nursing, which results in differentiation in terms of the time of professional experience, can therefore be used to establish the on-duty nursing staff, who will complement each other in terms of their level of knowledge and attitudes [30].

The development of nursing preparation to provide care for patients with sepsis by strengthening their knowledge and attitudes is most important in wards where it appears most often, i.e., intensive care units, emergency departments, pediatrics units, and Primary Care. Our study revealed that the highest level of knowledge is presented by nurses in intensive care units, emergency departments, and pediatrics units, which, hypothetically in relation to earlier results, might stem from the fact that, in their practice, they most often care for patients with sepsis. For example, according to the WHO, 41% of all global sepsis cases in 2017 occurred in children under five years of age [1]. Nevertheless, considering the high percentage of patients with sepsis diagnosed in Primary Care, the average level of knowledge of sepsis in this group of nurses indicates the need to address additional training for this group. For instance, the American Association of Critical-Care Nurses (AACN) emphasizes the importance of early recognition and intervention in sepsis management, advocating for nurse-led initiatives like the “HALT Sepsis—Think Sepsis First!” program to improve patient outcomes. Additionally, simulation-based training has been shown to enhance nurses’ ability to recognize and respond to sepsis effectively, thereby improving patient care in critical settings [36]. The other initiative is the UK’s e-learning program on sepsis aims to educate healthcare professionals in community-based settings, emphasizing the early identification and management of sepsis in adults and children [37].

Verification of the level of knowledge and attitudes by validated tools should be undertaken in all medical groups, as cooperation is the key to decreasing mortality and morbidity from sepsis. As our study shows, there remains a need to strengthen basic education, including simulations, as well as postgraduate education, in the field of knowledge of sepsis due to existing knowledge gaps.

## 6. Limitations

The study has some limitations. Due to the online survey, we cannot assess the response rate. Given the use of convenience sampling and voluntary participation, selection bias cannot be excluded. Moreover, the self-assessment of knowledge is subject to individual perception and may not accurately reflect actual competence, introducing the possibility of response bias. The different ages of nurses could result in bias with regard to the correlation of particular factors with age. The study protocol did not include an in-depth exploration of nurses’ experiences in terms of their workplace and place of residence. Additionally, this was a local study, so the findings should be interpreted with caution. Further research is needed in diverse regions, as local forms and methods of nursing education may influence nurses’ knowledge about sepsis. Moreover, as a cross-sectional study, it provides only a snapshot of knowledge at a single point in time and does not allow for assessing changes over time or establishing causal relationships.

## 7. Conclusions

Nurses demonstrated a moderate level of sepsis-related knowledge and attitude, with significant differences depending on the source of education and clinical experience. Higher levels of both objective and self-assessed knowledge, as well as more positive attitudes toward sepsis care, were observed among nurses who had completed postgraduate training and those with direct clinical exposure to sepsis, particularly in high-acuity settings such as intensive care, emergency, and pediatric units. The study confirms that relying solely on undergraduate education is insufficient to prepare nurses for effective recognition and management of sepsis. Additionally, nurses’ self-assessed knowledge was positively associated with their attitude, suggesting that increasing knowledge may contribute to greater readiness to act in clinical situations involving sepsis.

These findings support the need to enhance the content and quality of sepsis education in both pre- and post-registration nursing programs. Strengthening continuous professional development, expanding access to postgraduate training, and incorporating simulation-based learning may help improve clinical competencies and decision-making in sepsis care. In addition, the findings highlight the need to foster a culture of continuous professional development and ensure access to reliable self-education resources for nurses. Regular use of validated assessment tools may further support the identification of educational needs and the evaluation of intervention outcomes in diverse care settings.

## Figures and Tables

**Figure 1 nursrep-15-00195-f001:**
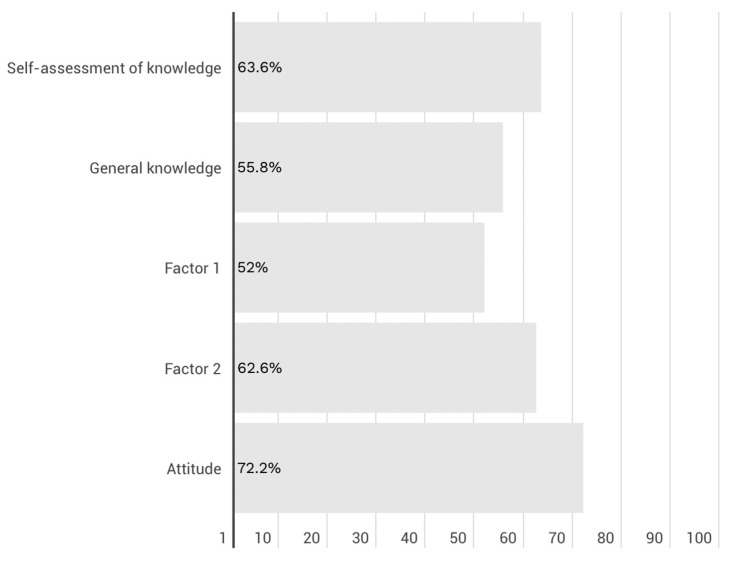
Percentage of nurses who provided correct answers in each dimension of the NAKSeS scale.

**Table 1 nursrep-15-00195-t001:** Sample characteristics.

	*n*	%
**Voluntary postgraduate education**		
Internal courses in the workplace	123	42
Specialization training	113	38.6
Qualification courses	153	52.2
Specialized courses	153	52.2
Other	25	8.5
None	39	13.3
**Current workplace**		
Department of Anesthesiology and Intensive Care	50	17.1
Emergency Ward	17	5.8
Primary Healthcare	43	14.7
Pediatric Ward (other than ICU)	17	5.8
Epidemiological nurse	2	0.7
Other	175	59.7
**Workplace**		
Village	16	5.5
City up to 50 thousand residents	56	19.1
City from 50 to 150 thousand residents	54	18.4
City from 150 to 500 thousand residents	66	22.5
City over 500 thousand residents	101	34.5
**The source of knowledge of sepsis**		
Undergraduate education	191	65.2
Postgraduate education	72	24.6
Professional experience	154	52.6
Books	102	34.8
Internet, TV, newspapers	90	30.7
Other	37	12.6

n = 293.

**Table 2 nursrep-15-00195-t002:** Pearson’s correlation coefficients between NAKSeS factors and self-assessed knowledge of sepsis.

	Self-Assessment of Knowledge	General Knowledge	Factor 1	Factor 2	Attitude
Self-assessment of knowledge	1				
General knowledge	0.18 **	1			
Factor 1	0.15 **	0.87 **	1		
Factor 2	0.13 *	0.68 **	0.24 **	1	
Attitude	0.70 **	0.21 **	0.15 **	0.20 **	1

*—*p* < 0.05, **—*p* < 0.001.

**Table 3 nursrep-15-00195-t003:** Comparison of the level of general knowledge, its dimensions, attitude and self-assessment of knowledge regarding the source of knowledge (professional experience and books).

	Other (n = 139)		Professional Experience (n = 154)				95% *CI*		
	*M*	*SD*	*M*	*SD*	*t*	*p*	*LL*	*UL*	*d*
General knowledge	8.71	3.30	10.19	3.31	−3.86	<0.001	−2.25	−0.73	0.45
Factor 1	5.24	2.45	6.14	2.57	−3.05	0.002	−1.48	−0.32	0.36
Factor 2	3.46	1.70	4.05	1.65	−3.02	0.003	−0.98	−0.21	0.35
Attitude	20.12	4.75	23.02	3.89	−5.68	<0.001	−3.90	−1.89	0.67
Self-assessment of knowledge	2.90	1.04	3.44	1.00	−4.49	<0.001	−0.77	−0.30	0.53
	**Other** **(n = 191)**		**Books** **(n = 102)**				**95% *CI***		
	*M*	*SD*	*M*	*SD*	*t*	*p*	*LL*	*UL*	*d*
General knowledge	8.96	3.36	10.47	3.20	−3.72	<0.001	−2.31	−0.71	0.46
Factor 1	5.43	2.59	6.25	2.39	−2.67	0.008	−1.43	−0.22	0.33
Factor 2	3.53	1.70	4.22	1.60	−3.33	0.001	−1.08	−0.28	0.41
Attitude	20.93	4.45	22.99	4.45	−3.78	<0.001	−3.14	−0.99	0.46
Self-assessment of knowledge	3.03	1.07	3.47	0.95	−3.51	0.001	−0.69	−0.20	0.43

**Table 4 nursrep-15-00195-t004:** Spearman’s correlations between general knowledge, factor 1, factor 2, attitude and self-assessment of knowledge and age and professional experience and workplace.

Variable	Age		Professional Experience (Years)		Workplace	
	*r_s_*	*p*	*r_s_*	*p*	*r_s_*	*p*
General knowledge	−0.12	0.042	−0.08	0.174	0.20	0.001
Factor 1	−0.18	0.003	−0.14	0.016	0.22	<0.001
Factor 2	0.04	0.516	0.06	0.308	0.03	0.602
Attitude	0.24	<0.001	0.27	<0.001	−0.07	0.242
Self-assessed knowledge of sepsis	0.13	0.023	0.15	0.009	−0.04	0.457

## Data Availability

Data are available on request from the authors.

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
