# Peer review of "Knowledge of Polish Nurses About Sepsis Based on Validated Questionnaire: A Multi-Site Cross-Sectional Study"

_nursrep, 2025, doi:10.3390/nursrep15060195_

Round 1
Reviewer 1 Report
Comments and Suggestions for Authors
Review of the manuscript titled: Knowledge of Polish nurses about sepsis based on validated questionnaire: A multi-site cross-sectional study
Submitted to: Nursing Reports
General Comments: This manuscript addresses a highly relevant and timely topic in nursing research—nurses’ knowledge of sepsis and its relationship with attitudes, self-assessment, and sociodemographic variables. Given the global burden of sepsis and nurses’ critical role in early detection, care, and monitoring, this topic has clear practical and scholarly significance. The study is well-structured, with an appropriate methodology and a valid instrument; however, the manuscript requires substantial revision to improve clarity, coherence, and presentation quality, particularly in the Introduction, Methods, and Results sections.
Abstract
The abstract in the Methods and Results section requires significant modification. You will find an explanation in the text below.
Introduction
The current introduction provides an extensive definition of sepsis, which is important, but it lacks focus on the research problem – nurses’ knowledge and attitudes toward sepsis.. Therefore, I strongly recommend focusing on previous research on nurses’ knowledge and attitudes regarding sepsis. What do they know well, what is insufficient, and what does their level of knowledge and attitudes depend on? Also, the introduction could mention how knowledge and attitudes are measured. State whether standardised questionnaires are used for this purpose or not. In addition, it would be good to provide data on the prevalence of sepsis in hospitals in Poland and how nurses are trained to monitor and provide care to these patients.
This will help to establish a clearer rationale and research gap for the study.
The study
Sample/Participants
The subtitle can be simplified to “Participants” or “Sample” – having both is redundant. I recommend that you opt for one term. In this part, for example, you use the terms subjects, participants, respondents and nurses; it also burdens the text. I recommend using the term nurses.
Please justify the criterion requiring a minimum of three months of work experience. Why was this chosen as the threshold?
Also, I’m not sure the term “didactic area” is adequate. Maybe education?
Data collection
Please, modify the sentence “Data collection was conducted by using the NAKSeS (Nurses Attitudes and Knowledge about Sepsis Scale) questionnaire and metrics developed by the authors of this study” – it sounds awkward. Namely, it is unnecessary to use multiple terms: scales, questionnaire and metrics. It is enough to write “Data collection was conducted using the NAKSeS (Nurses Attitudes and Knowledge about Sepsis Scale), which the authors of this study developed”.
The next awkward sentence - “The questionnaire contained a 23-item self-reported scale”. Please, modify it.
Describing the measuring instrument used in more detail and systematically is necessary. For example, how many correct and incorrect answers were in total from the knowledge assessment items? Were the attitude items all positively oriented, or were there some negative ones? Was reverse scoring necessary? What were the items in the self-assessment? ​​It is not clear.
This information is crucial for evaluating the instrument’s structure and interpretability.
Validity and reliability/Rigour
Choose “Validity” or “Rigour” for the section heading, not both.
Provide a reference for the reported reliability values of the original scale: “The reliability of the knowledge subscale in the original scale was KR-20 = 0.718, while Cronbach’s alpha for the attitude subscale was 0.884.”
Results/Findings
There is no need for both terms in the subtitle - opt ​​for one.
Again, select either “Results” or “Findings” for the section title.
I recommend that you opt for one term. In this part, for example, you use the terms subjects, participants, respondents and nurses; it also burdens the text. I recommend using the term nurses.
The results are presented in three tables and one figure. Display of results requires complete modification. Namely, a good statistical analysis was done, qualitative findings were obtained, and it is necessary to present them in tabular form, not just textually. Such results could be presented with two or three more tables.
The title of Figure 1 needs modification. The title should state that the percentage indicates the correct answer.
It is not easy to follow the results. For example, when reporting average scores (e.g., knowledge scores), include the scale range (e.g., “5.2 out of a maximum of 10”) to give meaning to the values. Results in sections 4.3, 4.4, .4.5, 4.6 and 4.7 should be presented in a table for clarity..
Discussion
The discussion addresses some, but not all, of the study’s findings. A more systematic explanation and comparison with existing literature is needed.
Limitations
This section is appropriately written. No major revisions are needed.
Conclusion
The conclusion is general. Therefore, the authors should explicitly address the conclusions for the aims, as this would provide a more comprehensive summary of the study’s outcomes.
Overall Assessment
The manuscript addresses a highly important issue in contemporary nursing practice and education. It has a sound structure and a well-designed methodology. However, substantial revisions are required, particularly in the introduction, instrument description, and results presentation. With these improvements, the manuscript has the potential to make a valuable contribution to the literature on sepsis awareness and nursing education.
Quality of English Language
I believe proofreading in English is also needed to improve the readability of the text. Authors often use strange or unscientific terms or inappropriate terms for academic writing.
Recommendation: Major Revision

I believe proofreading in English is also needed to improve the readability of the text. Authors often use strange or unscientific terms or inappropriate terms for academic writing.
Author Response
Dear Reviewer,
We would like to express our sincere gratitude for your careful reading of our manuscript and your insightful comments. Your feedback has been instrumental in refining and strengthening our work.
Below, we provide a detailed, point-by-point response to each of your comments. All changes made to the manuscript are clearly indicated and explained accordingly. We hope that the revised version adequately addresses your concerns and suggestions.
Thank you again for your time and valuable input.
Sincerely,
Authors

Reviewer 2 Report
Comments and Suggestions for Authors
This article addresses a crucial topic—nurses’ knowledge and attitudes toward sepsis—through a well-structured cross-sectional design using a validated tool (NAKSeS). The study is relevant, particularly in light of WHO priorities regarding sepsis awareness.
Strengths:
- The sample size is adequate and statistically justified.
- The questionnaire used is validated and well described.
- The study provides nuanced insights into the relationship between knowledge, attitudes, self-assessment, and demographic/professional variables.
Areas for Improvement:
- Methods:
- The presentation of the Likert scale choices (particularly for the attitude component) is somewhat confusing and inconsistently described in the text. Please clarify the order and label consistency.
- It would be useful to provide more detail about the dissemination of the questionnaire, particularly on how bias was mitigated given the convenience sampling strategy.
- Consider discussing response bias due to self-assessment in more depth in the limitations section.
- English Language:
- Several grammatical and syntactic errors reduce clarity (e.g., “the higher the score, the higher the level of knowledge” could be rephrased to avoid repetition).
- Improve consistency in terminology (e.g., “sepsis knowledge,” “knowledge of sepsis,” “general knowledge”).
- Consider professional language editing before publication.
- Figures and Tables:
- Figure 1 is referenced but not visible in the current version. Ensure all visuals are included and clearly labeled in the final manuscript.
- Table captions could benefit from more descriptive summaries of what the data represents (e.g., “Table 2: Pearson’s correlation coefficients between NAKSeS factors and self-assessed knowledge”).
- Discussion:
- The discussion is rich and evidence-based but could benefit from more critical engagement with international studies, including potential differences in nursing education systems.
- Emphasize more explicitly how your findings could inform nursing curricula or continuing education frameworks in Poland or more broadly
Author Response
Dear Reviewer,
Thank you very much for your thoughtful and constructive feedback on our manuscript. We sincerely appreciate your time, effort, and valuable suggestions, which have helped us to improve the quality and clarity of our work.
Below, we provide a detailed, point-by-point response to each of your comments. Changes made in the manuscript are clearly indicated and explained where applicable. We hope that the revisions address your concerns and meet the expectations for the manuscript.
Sincerely,
Authors

Round 2
Reviewer 1 Report
Comments and Suggestions for Authors
Dear Authors,
Thank you for your careful and thoughtful revisions to the manuscript. I appreciate your efforts in addressing my comments, which have significantly improved its overall quality.
You have successfully revised key sections of the manuscript, including the abstract, introduction, methodology, results, discussion, and conclusion, to align with the suggested recommendations. The manuscript is now clearer and more coherent and aligns with the standards of Nursing Reports.
I support the publication of this revised version and commend your multi-site cross-sectional research on Polish nurses’ knowledge about sepsis using a validated questionnaire.
Sincerely,
Reviewer